# Re-Sensitization of Resistant Ovarian Cancer SKOV3/CDDP Cells to Cisplatin by Curcumin Pre-Treatment

**DOI:** 10.3390/ijms26020799

**Published:** 2025-01-18

**Authors:** Aseel Ali Hasan, Elena Kalinina, Dmitry Zhdanov, Yulia Volodina, Victor Tatarskiy

**Affiliations:** 1T.T. Berezov Department of Biochemistry, Peoples’ Friendship University of Russia (RUDN University), 6 Miklukho-Maklaya Street, 117198 Moscow, Russia; kalinina-ev@rudn.ru (E.K.); zhdanovdd@gmail.com (D.Z.); 2Laboratory of Medical Biotechnology, Institute of Biomedical Chemistry, Pogodinskaya St. 10/8, 119121 Moscow, Russia; 3Laboratory of Tumor Cell Death, Blokhin National Medical Research Center of Oncology, 24 Kashirskoye Shosse, 115478 Moscow, Russia; uvo2003@mail.ru; 4Laboratory of Molecular Oncobiology, Institute of Gene Biology, Russian Academy of Sciences, 34/5 Vavilov Street, 119334 Moscow, Russia; tatarskii@gmail.com; 5Center for Precision Genome Editing and Genetic Technologies for Biomedicine, Institute of Gene Biology, Russian Academy of Science, 34/5 Vavilov Street, 119334 Moscow, Russia

**Keywords:** ovarian cancer, curcumin, cisplatin, thioredoxin reductase, apoptosis, signaling pathway

## Abstract

A major challenging problem facing effective ovarian cancer therapy is cisplatin resistance. Re-sensitization of cisplatin-resistant ovarian cancer cells to cisplatin (CDDP) has become a critical issue. Curcumin (CUR), the most abundant dietary polyphenolic curcuminoids derived from turmeric (*Curcuma longa*), has achieved previously significant anti-cancer effects against human ovarian adenocarcinoma SKOV-3/CDDP cisplatin-resistant cells by inhibition the gene expression of the antioxidant enzymes (*SOD1*, *SOD2*, *GPX1*, *CAT* and *HO1*), transcription factor *NFE2L2* and signaling pathway (*PIK3CA*/*AKT1*/*MTOR*). However, the detailed mechanisms of curcumin-mediated re-sensitization to cisplatin in SKOV-3/CDDP cells still need further exploration. Here, a suggested curcumin pre-treatment therapeutic strategy has been evaluated to effectively overcome cisplatin-resistant ovarian cancer SKOV-3/CDDP and to improve our understanding of the mechanisms behind cisplatin resistance. The findings of the present study suggest that the curcumin pre-treatment significantly exhibited cytotoxic effects and inhibited the proliferation of the SKOV-3/CDDP cell line compared to the simultaneous addition of drugs. Precisely, apoptosis induced by curcumin pre-treatment in SKOV-3/CDDP cells is mediated by mitochondrial apoptotic pathway (cleaved caspases 9, 3 and cleaved PARP) activation as well as by inhibition of thioredoxin reductase (TRXR1) and mTOR/STAT3 signaling pathway. This current study could deepen our understanding of the anticancer mechanism of CUR pre-treatment, which not only facilitates the re-sensitization of ovarian cancer cells to cisplatin but may lead to the development of targeted and effective therapeutics to eradicate SKOV-3/CDDP cancer cells.

## 1. Introduction

Ovarian cancer is the most popular malignant gynecologic cancer-related death among women [1]. The overall five-year survival rate is less than half of diagnosed women due to the predominance of aggressive high-grade serous carcinomas and the absence of early symptoms [2]. Significant risk factors, including stage, grade, and lymph node involvement, play a role in the development of distant metastases during ovarian carcinoma, which causes the disease. It can spread to various sites, such as the liver, pleura, and lung, and the survival rate of patients is based on their metastatic status [3]. Early detection and diagnosis of ovarian cancer is a possible way to improve survival rate by 10–30% [4].

The standard ovarian cancer therapy involves cytoreductive surgery followed by adjuvant platinum-based chemotherapy drugs, including cisplatin (CDDP) [5]. The anticancer activity of CDDP is mediated by a multidirectional mechanism that activates apoptosis processes in cells via damaging the DNA by formation of CDDP–DNA adducts, induction of oxidative stress and damaging the mitochondria of cancer cells. However, the mechanisms of CDDP resistance developed by ovarian cancer cells, including alterations in CDDP cellular transports, changes in the DNA damage repair mechanisms, as well as numerous changes in the apoptosis and autophagy processes, are emerging critical issues during therapy [6]. Accumulating data have shown that the transcription factor signal transducer and activator of transcription 3 (STAT3) that coordinates cancer cell survival, growth, progression, and CDDP resistance is activated in numerous types of cancer [7]. Ongoing research has aimed at developing reversal strategies for the treatment of CDDP-resistance ovarian cancer either by creating CDDP analogs that would be less toxic or using CDDP combined with radiotherapy or plant-derived substances [6].

Curcumin (CUR) [1,7-bis (4-hydroxy-3-methoxyphenyl)-1,6 heptadiene-3,5-dione], is the natural yellow pigment polyphenol curcuminoid compound, and the major chemical ingredient of turmeric that is derived from *Curcuma longa* [8]. CUR has many biological activities and potential health benefits, like antioxidants, anti-inflammatory, immune-regulatory activities, and anticancer effects [9]. The chemical structure of CUR consists of three different chemical entities: two aromatic ring systems containing o-methoxy phenolic groups, connected by a seven-carbon linker comprising an α, β-unsaturated β-diketone moiety [10,11]. Based on its β-diketone moiety, CUR exhibits keto-enol tautomeric forms, which can interact and bind with a wide range of enzymes [12]. The keto-enol-enolate equilibrium of the heptadienone moiety of CUR plays a crucial role in the physicochemical and antioxidant properties [13]. The keto form, through which CUR exerts its antioxidant activity, predominates in a polar/acidic medium. On the contrary, the enol form preponderates in a non-polar/alkaline medium [14].

CUR exerts its antioxidant activity that can significantly reduce oxidative stress by directing the removal of reactive oxygen and nitrogen, chelation of metal ions, and regulation of antioxidant-related enzyme activity [15]. CUR can also irreversibly inhibit both cytosolic/nucleus and mitochondrial thioredoxin reductase (TRXR1 and TRXR2, respectively) isoenzymes. TRXRs are essential mammalian selenocysteine (Sec)-containing flavoenzymes, which catalyze NADPH-dependent reduction of the active site disulfide in thioredoxin, that serves a wide range of functions in substrate reductions, defense against oxidative stress, cellular proliferation, and redox control [16]. A growing body of evidence has proved the anticancer effect of CUR on various cancers, including prostate cancer, cervical cancer, colorectal carcinoma, leukemia, and human breast cancer cells [17], which is mediated by inducing apoptosis and inhibiting the proliferation, invasion, and migration of cancers [18].

Studies show that CUR has the ability to inhibit ovarian cancer cells by promoting cell apoptosis, suppressing cell cycle progression, inducing autophagy and inhibiting tumor metastasis and invasion, as well as increasing their sensitivity to CDDP [19]. Moreover, CUR downregulated the expression of the Bcl-2 (Bcl-XL and Mcl-1) family of pro-survival proteins after 6 h, whereas CUR pre-treatment followed by exposure to low doses of CDDP increased the effectiveness of CDDP treatment in CDDP-resistant A2780CP ovarian cancer cells by increasing the sensitivity of cells to apoptotic pathways and modulating nuclear β-catenin signaling [20]. HO-3867, a curcumin analog, used in combination with CDDP against CDDP-resistant human ovarian cancer (A2780R) cell line, has a synergistic efficacy with significant induction of cell cycle arrest, apoptosis, and tumor growth inhibition through STAT3 inhibition [21].

We have previously shown that treatment with CUR alone demonstrates an anticancer effect against human ovarian adenocarcinoma SKOV-3/CDDP CDDP-resistant cells by inhibiting the gene expression of the antioxidant enzymes (*SOD1*, *SOD2*, *GPX1*, *CAT*, and *HO1*), transcription factor *NFE2L2*, and signaling pathway (*PIK3CA*/*AKT1*/*MTOR*) [22]. We have also demonstrated that the pre-treatment with quercetin (QU), a polyphenol compound, re-sensitized SKOV-3/CDDP cells to CDDP through induction of ROS production and apoptosis by the downregulation of the thioredoxin antioxidant system and mTOR/STAT3 signaling pathway [23].

The present study explores the effects of the CUR pre-treatment strategy on ovarian cancer SKOV-3/CDDP and its underlying mechanism. Our data supported that CUR pre-treatment suppressed the growth of ovarian SKOV-3/CDDP cancer cells. We also provided the evidence that CUR pre-treatment exhibits a significant induction of mitochondrial apoptotic pathway (cleaved caspases 9, 3 and cleaved PARP) in CDDP-resistant ovarian cancer SKOV-3/CDDP with therapeutic potential apparently due to resensitize to CDDP toxicity through downregulation of thioredoxin reductase (TRXR1) and mTOR/STAT3 signaling pathway. Here, we provided a better understanding of the molecular mechanisms underlying the re-sensitization of ovarian cancer SKOV-3/CDDP cells to CDDP by CUR pre-treatment.

## 2. Results

### 2.1. Combination CUR and CDDP Has an Antagonistic Cytotoxic Effect on SKOV-3 and SKOV-3/CDDP Cell Lines, While CUR Pre-Treatment Has a Synergistic Cytotoxic Effect Only on SKOV-3/CDDP Cells

The synergistic effects of the combination of CUR-CDDP and pre-treatment methods on both ovarian cells were assayed by the calculation of the combination index (CI) based on the Chou–Talalay method [24]. The IC_50_ of CUR and CDDP alone was reported previously [22]. Based on the IC_50_ value indicated after treatment with CUR or CDDP alone, both ovarian cancer cells were treated with a combination of CUR-CDDP at a constant ratio (1:1), and the effect was studied after 72 h of treatment. Different concentrations of CUR were employed firstly for 24 h and then followed by different concentrations of CDDP at a constant ratio (1:1) for another 72 h after replacing the medium with a new one.

Figure 1 shows the dose-effect curves of CUR, CDDP, CUR-CDDP combination (Figure 1A) and CUR pre-treatment (Figure 1B), and the combination index values of the CUR-CDDP combination and CUR-pre-treatment obtained using CompuSyn calculations (Figure 1C). The CUR-CDDP combination showed high antagonistic effect (CI > 1) toward both SKOV-3 and SKOV-3/CDDP cell lines (Figure 1A,C). An alternative pre-treatment method showed antagonistic effect (CI > 1) in SKOV-3 cells (Figure 1B,C), while in SKOV-3/CDDP cells, a pre-treatment resulted in an effective synergistic effect (CI < 1) (Figure 1B,C).

The CI values of all combinations and pre-treatments at different inhibition growth in both ovarian cells are shown in Figure 1. The CUR-CDDP combination showed no efficacy in inhibiting the growth of both SKOV-3 and SKOV-3/CDDP cancer cell lines compared to CUR or CDDP alone. These results indicated that combining CUR and CDDP has no anticancer effect against both ovarian cancer cells (Figure 1A,C). The pre-treatment method has a significant synergistic anticancer effect against only SKOV-3/CDDP ovarian cancer cells (Figure 1B,C).

### 2.2. CUR Pre-Treatment Has a Synergistic Cytotoxic Effect in the SKOV-3/CDDP Cell Line

To determine whether CUR sensitizes the SKOV-3/CDDP cell line to CDDP, a suggested CUR pre-treatment therapeutic strategy was designed (Figure 2A). To ensure that CUR could sensitize SKOV-3/CDDP to CDDP at low doses, both cell lines were incubated with 17 μM CUR for 24 h, then the medium was replaced with a fresh culture medium and incubated for another 72 h with different concentrations of CDDP, resulting in enhancement the sensitivity of both ovarian cell lines to CDDP and cell growth inhibition compared to each drug alone (Figure 2B,C).

Treatment of SKOV-3/CDDP and SKOV3 cell lines with CUR or CDDP alone was found to be ineffective in inducing the inhibition of growth, as the percentage of viability was higher than 70%. However, this treatment became highly effective (< 2µM) when both cell lines were pre-treated with CUR followed by CDDP (Figure 2B,C).

### 2.3. CUR Pre-Treatment Modulates Cell Cycle Distribution in SKOV-3 and SKOV-3/CDDP Cell Lines

To examine the mechanism involved in CUR pre-treatment mediated cell growth inhibition, cell cycle phase distribution of cells treated with either CUR or CDDP and the CUR pre-treatment for 24 h, 48 h, and 72 h was analyzed by flow cytometry using a propidium iodide staining assay. As shown in Figure 3A,B and Appendix A, CDDP alone induced S phase arrest in both cell lines, while treatment of both ovarian cancer cells with CUR alone resulted in >35% of the cells arrested in the G2/M phase of the cell cycle by 24 h. This block was resolved in SKOV-3 cells at 24 h, 48 h, and 72 h after the replacement of the medium, as well as an increase in the G0/G1 phase.

The sub-G1 phase, marker for apoptotic cells, significantly increased only in CUR pre-treatment (to >40%) in both SKOV-3 and SKOV-3/CDDP cell line cells compared to CUR or CDDP groups at 72 h (Figure 3C,D). For the SKOV-3/CDDP cell line, there was little change in survival until approximately 48 h.

Our study proved that CUR pre-treatment increased the CDDP activity compared with combination treatment. Cell cycle results showed that the combination of CUR with CDDP when both drugs were applied at the same time had no additive or synergistic effects on apoptosis in SKOV-3/CDDP cells, without an increase in subG1 cells after long incubation (for 72 h) (Appendix A).

### 2.4. CUR Pre-Treatment Induces Apoptosis in SKOV-3/CDDP Cell Line

To further confirm the mode of death in CUR pre-treatment cells, double staining Annexin V FITC and PI followed by flow cytometry analysis was used. As shown in Figure 4A,B, no significant increase in total apoptosis was observed in ovarian SKOV-3/CDDP cancer cell lines treated with either agent alone compared to untreated cells. However, the CUR pre-treatment increased the percentage of early and total apoptotic cells to 13.5% and 42% at 72 h, respectively (Figure 4A–D).

Furthermore, the expression of apoptosis-related proteins, including cleaved caspase 9, cleaved caspase 3, and cleaved PARP in SKOV-3/CDDP cells, was evaluated by western blot analysis. As shown in Figure 4E, treatment with either CUR or CDDP alone had no significant effect on the protein levels of the apoptosis-related proteins in the cell line. The pro-apoptotic protein levels of cleaved caspase 9, cleaved caspase 3, and cleaved PARP were higher in SKOV-3/CDDP cells subjected to CUR pre-treatment than in untreated control cells and cells treated with either agent alone (Figure 4E). These results indicated that CUR pre-treatment exerted an anti-proliferative effect in ovarian SKOV-3/CDDP cancer cells by inducing apoptotic pathways.

### 2.5. High Expression of TXNRD1 as Well Phosphorylated mTOR Level Correlate with Resistance to CDDP in Cancer Cell Lines

We previously showed that CDDP-resistant SKOV-3/CDDP cells have increased levels of TRXR1, phosphorylated mTOR and phosphorylated STAT3 compared to SKOV-3 cells [23]. To determine whether overexpression of thioredoxin reductase, phosphorylated-mTOR and STAT3 in SKOV-3/CDDP leads to cellular CDDP resistance in other cell lines, we utilized the DEPMAP database to verify these results. The database contains data for sensitivity to CDDP treatment and gene expression and protein levels, for 496 cell lines. These data were used to correlate chemosensitivity data of CDDP and transcriptome-level gene expression/RPPA protein levels.

As shown in Figure 5, analysis of cell lines in the DEPMAP showed that high levels (>median) of the *TXNRD1* gene and increased levels of mTOR p-Ser^2448^ protein are statistically significantly correlated to resistance against CDDP in tested cancer cell lines. The number of cancer cell lines with low expression of the *TXNRD1* gene and mTOR p-Ser^2448^ protein that show a sensitivity (AUC < 0.9) to CDDP (29.9% and 24.03%) was higher compared to the cell lines with high expression of TRXR1 and mTOR p-Ser^2448^ (13.09% and 15.12% of cell lines) that are sensitive to CDDP. Although the number of resistant cell lines with increased levels of *STAT3* was higher, this difference was not statistically significant. 

### 2.6. High Expression of Thioredoxin System and Peroxiredoxins Genes Could Lead to CDDP Resistance in SKOV-3/CDDP

Previously, we demonstrated the gene expression of antioxidant enzymes (*SOD1*, *SOD2*, *GPX1*, *CAT* and *HO1*), transcription factor *NFE2L2*, and signaling pathway (*PIK3CA*/*AKT1*/*MTOR*) [22] and protein expressions of the thioredoxin system and mTOR/STAT3 signaling proteins [23] were higher in SKOV-3/CDDP cells than in SKOV-3 cells. Overexpression of these genes could contribute to acquiring resistance to CDDP in SKOV-3/CDDP cells. Here, we have screened thioredoxin system genes that are known to be associated with CDDP resistance in SKOV-3/CDDP cells by using real-time quantitative RT-PCR. As presented in Figure 6, SKOV-3/CDDP cells showed increased expression of the thioredoxin system compared to SKOV-3 cells. These findings suggested that increased expression of the thioredoxin system is associated with CDDP resistance in ovarian SKOV-3/CDDP cancer cells.

### 2.7. CUR Pre-Treatment Inhibits TRXR1, mTOR, and STAT3 in the SKOV-3/CDDP Cell Line

To explore the potential relationship between the thioredoxin system and mTOR/STAT3 activity and CUR/CDDP responsiveness in SKOV-3/CDDP cells, we analyzed the expression of these proteins by western blot analysis. Western blot assay was used to assess the protein levels of TRXR1, P-mTOR and P-STAT3 to verify the data of mRNA expression. We previously proved that increased levels of TRXR1 phosphorylated mTOR and phosphorylated STAT3 in SKOV-3/CDDP cells compared with SKOV-3 cells [23]. CUR pre-treatment effectively reduced the levels of TRXR1, mTOR p-S^2448^, STAT3 and STAT3 p-S^727^ in SKOV-3/CDDP cells as comparable to cells treated with CUR or CDDP alone (Figure 7). These results indicate that CUR pre-treatment could exhibit anti-cancer effects via inhibiting TRXR1 and mTOR/STAT3 signaling pathways.

### 2.8. Intracellular CUR Accumulation at Different Cell Densities

Intracellular CUR accumulation at different cell densities was assessed by flow cytometry. As shown in Appendix A, the maximum mean fluorescence of CUR accumulation was 1855 at lower seeding density (15,625/cm^2^) in 6-well plate, while the fluorescence accumulation sharply decreased to 930 and 913 at high densities (42,857/cm^2^ and 78,947/cm^2^) in 12 and 24-well plates, respectively. We observed that the cytotoxicity assay and cell death variations were correlated with density-dependent variations in the SKOV-3/CDDP cell line. Thus, the effect of CUR is inversely related to the seeding density in SKOV-3/CDDP cells. To minimize these variations, the cell seeding density was optimized for all experiments, as follows: the MTT assay was 5 × 10^3^ per well in a 96-well plate and further experiments were 1.5 × 10^5^ per well in a 6-well plate.

## 3. Discussion

Cisplatin (CDDP) is primarily considered an anticancer drug for ovarian cancer treatment. However, acquired cisplatin (CDDP) resistance can be attributed to changes in the activity of protein transporters, an increase in the effectiveness of DNA damage repair mechanisms and a change in the processes of apoptosis and autophagy in CDDP-resistant ovarian cancer cells. New strategic approaches for ovarian cancer treatment are intensively sought [6]. The combination of CDDP and natural compounds is gradually becoming an important strategy to alleviate CDDP resistance and to develop a direction for human cancer treatment [25]. Notable among these natural compounds is curcumin (CUR) as a natural polyphenol compound, which exhibits antioxidant, anti-inflammatory and anticarcinogenic properties. The anti-tumor activities of CUR include inhibition of ovarian cancer cell proliferation, invasion and metastasis, induction of cell apoptosis and autophagy, as well as the increase of chemotherapy sensitivity [19]. Despite extensive studies that have shown the synergistic role of CUR in enhancing the efficacy of the most common chemotherapy drugs, its potential therapeutic role in ovarian cancer still needs to be elucidated [26].

Based on the preclinical studies, polyphenols have been shown to work synergistically with chemotherapeutic drugs, but some polyphenols work antagonistically [27]. Unexpectedly, in the current study, CUR, when used with CDDP, shows an antagonistic effect in constant ratio (1:1) (Figure 1). In the CDDP-sensitive SKOV-3 cell line and CDDP-resistant SKOV-3/CDDP cell line, the CUR-CDDP combination was not able to inhibit cell proliferation and induce cytotoxicity compared to a single treatment (Figure 1A). CUR pre-treatment for 6 h effectively sensitized CDDP-resistant A2780CP ovarian cancer cells to the cytotoxic effects of CDDP [20]. To improve our understanding of the mechanisms underlying CDDP resistance, we used an alternative CUR pre-treatment therapeutic strategy in our study. Pre-treatment methods effectively had a strong synergistic effect at a constant ratio (1:1) against SKOV-3/CDDP cells (Figure 1B). Our study’s cytotoxicity results showed that pre-treatment ovarian SKOV-3 and SKOV-3/CDDP cancer cells with CUR followed by CDDP significantly inhibited cell viability in a dose-dependent manner compared to CUR or CDDP treatment alone (Figure 2). These data strongly suggest the possible reduction in cell viability observed in the MTT assay may be caused by the inhibitory effect of CUR pre-treatment on ovarian cancer cell proliferation.

Furthermore, the CUR pre-treatment could reduce the proliferation and eliminate the ovarian cancer cells in a time-dependent manner. CUR has been shown to induce G2/M phase cell-cycle arrest in CDDP-resistant (CR) human ovarian cancer cells by enhancing apoptosis through the activation of caspase-3 followed by PARP degradation [28]. As shown in Figure 3 and Appendix A, CUR induced a cell cycle arrest in G2/M in both cells after 24 h and induced arrest in either G0/G1 (SKOV3) or G2/M (SKOV3/CDDP) after replacement of medium, while CDDP induced a cell cycle arrest in the S phase in SKOV-3 cells and in S and G2/M phases in SKOV-3/CDDP. CUR pre-treatment efficiently induced the accumulation of both ovarian cells in the sub-G1 phase by more than 40% of the total cell population after 72 h of exposure, indicating effective cell death induction.

The data from CompuSyn were confirmed at the cellular level. The combination of CUR with CDDP when both drugs were applied simultaneously had no additive or synergistic effects on apoptosis in SKOV-3/CDDP cells, without an increase in subG1 cells after long incubation of 72 h (Appendix A). Due to strong antioxidative activities, polyphenols could prevent cisplatin-induced damage and prevent acute kidney injury by scavenging ROS [29]. For example, green tea polyphenol, epicatechin gallate (ECG), prevented cisplatin-induced oxidative stress, inflammation, and apoptosis by downregulating the MAPK pathway, leading to improved renal function in cisplatin-treated rats [30]. The antagonistic effect can be explained by one drug targeting cells at different phases of the cell cycle, when one drug prevents the progression of the cell cycle to a phase where the second drug has its effect, making the combination treatment less active than mono-treatment [31]. In our case, blocking of the cells at the G2/M phase by CUR, can prevent further damage in the replication phase by CDDP, while the washout of the drug increases the susceptibility of damaged cells to the CDDP in the subsequent round of replication.

Flow cytometry (annexin V/PI staining) confirmed the reduction in the number of viable cells is also caused by induction of apoptotic SKOV-3/CDDP cancer cell death. CUR pre-treatment in SKOV-3/CDDP significantly induced apoptotic cell death compared to CUR or CDDP treatment groups (Figure 4A–D). These results were confirmed at the protein level by western blot analysis (Figure 4E), CUR pre-treatment suppressed SKOV-3/CDDP cell survival via the induction of caspase-mediated apoptosis through the activation of caspase-3 followed by PARP degradation (Figure 4E).

Activation of nuclear factor erythroid 2-related factor 2 (NRF2), as a transcription factor, has been proposed to contribute to chemo-resistance in cancer cells. NRF2 controls the expression of several enzymatic antioxidants (*SOD1*, *SOD2*, *GPX1*, *CAT*, and *HO1*), and results in the modulation of the levels of reactive oxygen species (ROS) [32]. Activation of PI3K-AKT-mTOR signaling and NRF2 make cancer cells resistant to oxidative stress through an enhanced antioxidant system, which in turn results in drug resistance [33]. Our previous findings indicated that overexpression of antioxidant enzymes (*SOD1*, *SOD2*, *GPX1*, *CAT*, and *HO1*), transcription factor *NFE2L2* and signaling pathway (*PIK3CA*/*AKT1*/*MTOR*). has been directly associated with resistance to CDDP in human ovarian adenocarcinoma SKOV-3/CDDP CDDP-resistant subline cells compared with the original SKOV-3 cell line. Treatment with CUR alone significantly inhibited the growth of human ovarian adenocarcinoma SKOV-3/CDDP CDDP-resistant subline cells by inhibition the gene expression of the antioxidant enzymes (*SOD1*, *SOD2*, *GPX1*, *CAT*, and *HO1*), transcription factor *NFE2L2* and signaling pathway (*PIK3CA*/*AKT1*/*MTOR*) [22]. Using genomic and proteomic data in DEPMAP database, we found that the expression of *TXNRD1* and *STAT3* genes and mTOR pS^2448^ protein were higher in CDDP resistance cancer cells compared to sensitive cancer cells to CDDP (Figure 5). The DEPMAP data were matched to our gene expression results that were obtained using real-time quantitative RT-PCR. SKOV-3/CDDP cells showed overexpression in thioredoxin system genes (Figure 6). The thioredoxin system is an antioxidant system that works to establish and maintain a reduced intracellular redox state. However, accumulating evidence suggests that the overexpression of the thioredoxin system indicates its possible involvement in the process of oncogenesis and contributes to resistance to therapy by alteration of molecular mechanisms and cell signaling pathways involved in the regulation of apoptosis in cancer cells [34].

Due to the thiol groups in the Cys residues in the active sites of thioredoxin, reduced thioredoxin participates in the reduction of oxidized target peroxiredoxins (Prxs) proteins [35], which are classified into three subtypes of typical 2-Cys, atypical 2- cysteine residues and 1- cysteine residue [36]. Thus, reduced Prxs detoxify ROS, e.g., H_2_O_2_ to water and oxygen [35,36]. In normal cells, Prxs have a significant effect on various physiological functions, including cell growth, differentiation, apoptosis, cellular homeostasis and redox signaling. Increasing evidence has hypothesized that overexpression of the Prx isoforms is responsible for the development of drug resistance as well as carcinogenesis by sustained ROS resistance in cancer cells [36]. For instance, Prx-2 overexpression contributes to chemoresistance by inhibiting cisplatin-induced apoptosis in SNU638 cells [37]. A high level of Prx-6 expression promotes a level of cisplatin resistance by enhancing stem-like properties in human lung cancer cells [38]. Here, it is confirmed that SKOV-3/CDDP subline cells exhibited overexpression of Prx mRNA (*PRDX-1,2,3,5,* and *6*) isoforms (Figure 6). The mammalian thioredoxin reductase isoenzymes, TRXR1 in cytosol or nucleus, TRXR2 in mitochondria, are selenocysteine (Sec)-containing flavoenzymes with a unique C-terminal -Gly-Cys-Sec-Gly active motif that only means of catalyzing reduction of thioredoxin. CUR showed irreversibly inhibition of TRXR [16].

These results were confirmed at the protein level using western blot analysis (Figure 7). We found that the expression of only TRXR1, mTOR pS^2448^, STAT3 and STAT3 pS^727^ proteins in ovarian SKOV-3/CDDP cancer cells treated with CUR followed by CDDP were downregulated, which led to induction of cell death in these cells (Figure 7). Our previous result has also shown that quercetin, a polyphenol compound, could repress SKOV-3/CDDP cell line growth and re-sensitize cells to CDDP through induction of ROS production and apoptosis by downregulation of thioredoxin antioxidant system and mTOR/STAT3 signaling pathway [23]. The reduced form of thioredoxin catalyzes the reduction of disulfide bonds in target proteins [39,40], including STAT3 [40]. Elevated levels of Thioredoxin reductase and STAT3 signaling have been associated with the development of chemotherapy resistance (Figure 5, Figure 6 and Figure 7). In cancer cells, the inhibition of TRXR1 expression results in increased oxidative stress and the accumulation of oxidized Prx2 and STAT3, which blocks STAT3-dependent transcription [41]. In this study, after CUR-CDDP treatment, the inhibition of TRXR1 after CUR-CDDP treatment results in the accumulation of oxidized STAT3, which blocks STAT3-dependent transcription. Targeting STAT3 signaling by CUR-CDDP could interrupt the anti-apoptotic function of ovarian cancer cells, which leads to increased apoptotic cell death. Furthermore, activation of mTORC1 leads to direct Ser^727^ phosphorylation of STAT3 during hypoxia and promotes hypoxia-inducible factors (HIF-1α) mRNA transcription and vascular endothelial growth factors (VEGF-A) [42]. Thus, our results suggested that CUR pre-treatment could inhibit mTOR phosphorylation. This, in turn, leads to a decrease of phosphorylation in Ser^727^ in STAT3 to confer re-sensitization to CDDP drugs in SKOV-3/CDDP cells. We observed that the behavior of a SKOV-3/CDDP cancer cell line could vary between chemoresistant and chemosensitive to CUR in an unpredictable manner depending on the seeding densities (Appendix A). 

Altogether, the lack of synergistic effect in combination strategy has made it ineffective to re-sensitize ovarian SKOV-3/CDDP cancer cells to CDDP. In contrast, the CUR pre-treatment represents a promising alternative method of therapy designed to boost the CDDP effect specifically ovarian CDDP- resistance SKOV-3/CDDP cancer cell line through induction apoptosis by down-regulating TRXR1, mTOR/STAT3.

## 4. Materials and Methods

### 4.1. Reagents and Drugs

Curcumin (CUR, (98% purity)) was obtained from (Acros Organics, Hong Kong, China). The anticancer drug cisplatin [CDDP, Cis-diamminedichloroplatinum (II)] was purchased from (Teva, Tel Aviv-Yafo, Israel). [3-(4,5-dimethylthiazol-2-yl)-2,5-diphenyl tetrazolium bromide] (MTT) was obtained from (PanEco, Moscow, Russia). The primary antibodies against mTOR, mTOR (pS2448), and TrxR-1 were purchased from Abcam (Cambridge, MA, USA). The primary antibodies against STAT3; STAT3 (pY705 and pS727) were purchased from Sigma-Aldrich (St. Louis, MO, USA). The primary antibodies against caspase 3; caspase 9, PARP; the cleaved caspase 3 (Asp175); cleaved caspase 9 (Asp330); cleaved PARP (Asp214), GAPDH (as reference protein), and the secondary antibodies labelled with horseradish peroxidase were purchased from Cell Signaling Technology, Inc., (Danvers, MA, USA). CUR was stored at −20 °C as a 10 mM stock solution in dimethyl sulfoxide (DMSO, PanEco, Moscow, Russia) and protected from light. The stock solution of CUR was diluted by the culture medium and used in indicated concentrations.

### 4.2. Cell Lines and Culture Conditions

Human ovarian cancer SKOV-3 and SKOV-3/CDDP cell lines were obtained from the All-Russian Scientific Center for Molecular Diagnostics and Treatment. These cells were maintained as monolayer cultures in DMEM cell culture medium supplemented with 10% heat-inactivated fetal bovine serum (FBS) (HealthCare, Chicago, IL, USA), 1% L-glutamine and 1% penicillin-streptomycin at 37 °C in a humidified atmosphere (5% CO_2_). All experiments were performed when the cells reached 85–90% confluence. All treatments were performed under non-cytotoxic conditions. IC_50_ values for CUR (17 µM and 47 µM for SKOV-3 and SKOV-CDDP, respectively) and CDDP (10 µM and 34 µM for SKOV-3 and SKOV-3/CDDP cells, respectively) were established in our previous papers [22,23]. Treatments with CDDP (½ IC_50_, 5µM, and 17µM for SKOV-3 and SKOV-3/CDDP cells, respectively) were performed alone for 24, 48, and 72 h, and CUR (IC_50_ for SKOV-3 cells) for 24 h followed by replacement with either only fresh medium for additional 24, 48 h, and 72 h, as mono CUR treatment or followed by (½ IC_50_, 5µM, and 17µM for SKOV-3 and SKOV-3/CDDP cells, respectively) CDDP treatment for an additional 24, 48, and 72 h after changing medium, as CUR pre-treatment. Each compound was investigated in three independent experiments; each experiment was carried out in triplicate.

### 4.3. Evaluation of Combination and Pre-Treatment Methods

The effect of CUR or CDDP alone on cell proliferation was assessed previously [22] using the MTT assay. To determine whether there is a synergistic effect between CUR and CDDP, two different treatment methods were tested in this study: combination and pre-treatment methods. The combination index (CI) was analyzed by using the Compusyn software program (Version 1.4) (http://www.combosyn.com/, accessed on 1 October 2024), which utilizes the Chou–Talalay equation method [24].

Based on pre-calculated IC_50_ values for each drug separately, SKOV-3 and SKOV-3/CDDP cells were treated with either a combination of CDDP-CUR at a constant ratio (1:1) for 72 h or were treated firstly with CUR for 24 h, then the culture medium was replaced with a fresh medium, and the cells were treated with CDDP at constant ratio (1:1) for another 72 h. The data obtained with the MTT assay was normalized to vehicle control. Furthermore, the data were converted to the fraction affected (Fa; range 0–1), where Fa = 0 represents 100% viability and Fa = 1 represents 0% viability). CI < 1 indicates synergism, CI = 1 indicates an additive effect, and CI > 1 indicates antagonism.

### 4.4. CUR Pre-Treatment Cytotoxicity Against SKOV-3 and SKOV-3/CDDP Cell Lines

In order to evaluate the efficacy of CUR sensitization prior to addition of CDDP treatment at fixed concentration, SKOV-3 and SKOV-3/CDDP cells were treated with optimum dose of (17 µM, IC_50_ for SKOV-3) CUR for 24 h based on previously published data [22], then the culture medium was replaced with fresh medium and the cells were treated with different CDDP concentrations 0.1–50 μM for another 72 h. After incubation, 5 mg/mL MTT reagent was added to each well, and the microplates were incubated for 4 h in a humidified atmosphere (37 °C, 5% CO_2_). DMSO (100 µL) was later added to the cells, and the absorbance at 570 nm was measured using an ELISA plate reader (Bio-Rad Laboratories, Hercules, CA, USA) using wells without cells as the blank. The cell viability of CUR pre-treatment in both cell lines was normalized to the CUR control cells treated with (17 µM/24h followed by fresh medium for 72 h) CUR alone and calculated according to the following formula: Cell viability (%) = cells (CUR pretreated sample)/cells (CUR control) × 100. The standard curve for each tested drug was constructed using an MTT assay. The data were visualized and analyzed by using the GraphPad Prism software version 8.0.2. (GraphPad Prism, La Jolla, CA, USA).

### 4.5. Cell Cycle Assay

Cell cycle distribution and ploidy status of cells after treatment with CDDP alone or CUR alone or followed by CDDP were determined by flow cytometry. At the end of treatments, cells were detached from the plates by the addition of trypsin, washed in phosphate-buffered saline (PBS) and treated with lysis buffer (containing 0.1% sodium citrate, 0.3% NP-40, 50 μg/mL RNAse A, and 10 μg/mL PI) for 30 min at 37 °C. The DNA content was evaluated in a BD FACS Canto II flow cytometer in a PE channel. For cell cycle analysis, only single cells were considered. A pass filter of 585/42 nm was used to collect PI fluorescence, acquiring 10,000 events for each sample.

### 4.6. Apoptotic Programmed Cell Death Analysis

To determine the effect of CUR pre-treatment on cell apoptosis, annexin V/PI staining (ApoFlowEx FITC Kit, Exbio, Czech Republic), was employed to detect the mode of death in ovarian SKOV-3/CDDP cancer cells. Briefly, the treated cells were collected, washed twice with PBS and suspended in 100 µL binding buffer. Annexin V/FITC solution was added to the cells, followed by the addition of 10 µL PI. Stained cells were then detected with a flow Cytoflex cytometer (BD Biosciences, Franklin Lakes, NJ, USA).

### 4.7. Data Collection and Drug (CDDP) Screening

To provide information on genes or protein overexpression that contribute to CDDP resistance, we used data from DEPMAP (Cancer Dependency Map). Drug (CDDP) sensitivity data (area under curve, AUC) for 494 cancer cells were obtained from GDSC2 (Genomics of Drug Sensitivity in Cancer) screening [43]. For gene expression data 24Q2 RNAseq build was used as a source [44], and for protein levels, protein array (RPPA) data from Cancer Cell Line Encyclopedia (CCLE) was used, both as provided by the DEPMAP portal.

### 4.8. RNA Isolation and Real-Time Quantitative RT-PCR

In order to verify the data obtained from the DEPMAP database, the mRNA expression levels of the candidate overexpressed genes associated with resistance to CDDP were measured by real-time quantitative RT-PCR. Total RNA from cells was extracted using the PureLink RNA Mini Kit (Thermo Fisher Scientific Inc., Waltham, MA, USA) according to the manufacturer’s protocol. Reverse transcription and real-time RT-PCR were performed as previously described [45]. In total, 5 µg of total RNA was reverse-transcribed using an MMLV RT kit (Evrogen, Moscow, Russia) in a 25-µL reaction mixture, followed by real-time RT-PCR using DTprime5 (DNA Technology, Protvino, Russia). The reaction mix was prepared using qPCRmix-HS SYBR according to the manufacturer’s recommendations using primers listed in Table 1. Two annealing/extension temperature cycles were used. The fluorescence was measured at the end of the annealing step. Melting curve analyses were performed at the end of the reaction (after the 45th cycle) between 60 °C and 95 °C to assess the quality of the final PCR products. The standard curves for reaction effectiveness were performed using 4 serially diluted samples (1:40, 1:80, 1:160, and 1:320). The relative expression levels of the target genes were normalized to the GAPDH internal control. The calculation of the relative RNA concentration was performed using the DTprime5 software (version 7.9).

### 4.9. Western Blotting Assay

To emphasize whether CUR pre-treatment could enhance the sensitivity of SKOV-3/CDDP to CDDP by suppressing the overexpressed genes, we evaluated protein levels by western blotting. Briefly, total protein was extracted from treated cells. The cells were detached and harvested in RIPA lysis buffer (50 mM Tris-HCl, pH 7.4, 1% NP-40, 0.25% Na-deoxycholate, 150 mM NaCl, 1 mM Na_3_VO_4_, and 1 mM NaF), then kept on ice for 30 min and centrifuged at 10,000× *g* for 15 min. Protein content was measured using the Coomassie Plus Bradford (Sigma) assay kit. Equal amounts of extracted proteins (40 mg/sample) were separated by 10 or 12% sodium dodecyl sulfate (SDS)–polyacrylamide gel electrophoresis under denaturing conditions and electroblotted onto 0.2 um nitrocellulose membranes (Bio-Rad, Hercules, CA, USA). The membranes were incubated with blocking buffer containing 5% non-fat dry milk in Tris-buffered saline Tween-20 (TBST) buffer (25 mM Tris–HCl, pH 7.5, 150 mM NaCl, 0.1% Tween-20) and then incubated with the primary antibody. The membranes were washed with PBS and incubated with the horseradish peroxidase-conjugated secondary antibody. The signals were visualized by chemiluminescent detection according to the manufacturer’s protocol Clarity Western ECL Substrate (Bio-Rad) using an iBright FL1500 Imaging System (Invitrogen, Waltham, MA, USA). The membranes were reprobed with anti-GAPDH antibodies to confirm an equal protein loading.

### 4.10. Intracellular Accumulation

To investigate the effects of seeding density on the MTT-measured IC50 values, cell cycle arrest, and apoptotic cell percentages of CUR, the fluorescence intensity of intracellular accumulation of CUR was assessed by flow cytometry. Briefly, the CDDP-resistant SKOV-3/CDDP cell line was seeded in 6-well, 12-well, and 24-well plates at a density of 1.5 × 10^5^ per well overnight. SKOV-3/CDDP cell line was then treated with 17 μM CUR for 24 h. Cells were then collected as a single-cell suspension and washed twice with ice-cold phosphate-buffered saline. The accumulation of CUR within cells was evaluated by the intracellular fluorescence intensity, using a BD FACSCanto II (BD Biosciences, San Jose, CA, USA) in the APC channel, acquiring 5000 events for each sample.

### 4.11. Statistical Analysis

All experiments were repeated at least three times, and the data were shown as mean ± standard deviation (SD). GraphPad Prism version 8.0.2 was used in statistical analysis. Significance was determined using ANOVA with Dunnett’s multiple comparisons correction. Meanwhile, the IC_50_ and combination index values were calculated using GraphPad Prism version 8.0.2 and synergy scores were calculated using Compusyn software. All of the Graphs were generated in GraphPad Prism version 8.0.2.

## Figures and Tables

**Figure 1 ijms-26-00799-f001:**
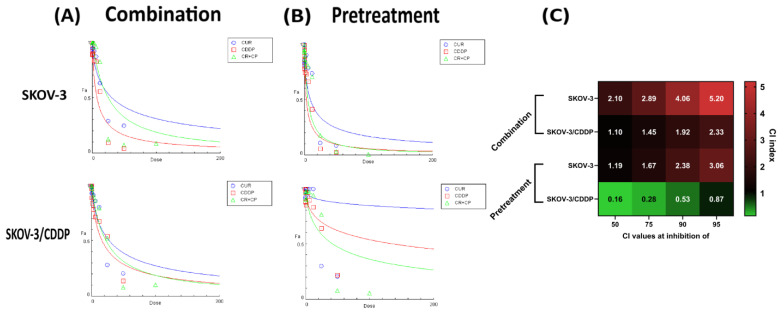
CompuSyn analysis of cytotoxicity data was used to determine synergy, additivity and antagonism between CUR and CDDP in SKOV-3 and SKOV-3/CDDP ovarian cancer cells after (**A**) combination and (**B**) pre-treatment schemes. Dose-effect curves of CUR, CDDP, and the CUR-CDDP combination at constant ratio [1:1] in SKOV-3 and SKOV-3/CDDP ovarian cancer cells. The *Y* axis, fraction affected (Fa) represents the ratio of the combination effect on inhibition of cell growth; the *X* axis represents the doses of drugs. (**C**) CI values for CUR and CDDP combination and pre-treatment at 50%, 75%, 90% and 95% inhibition of SKOV-3 and SKOV-3/CDDP ovarian cancer cell growth. All data were generated from the CompuSyn calculations, and the values are the mean of three experiments.

**Figure 2 ijms-26-00799-f002:**
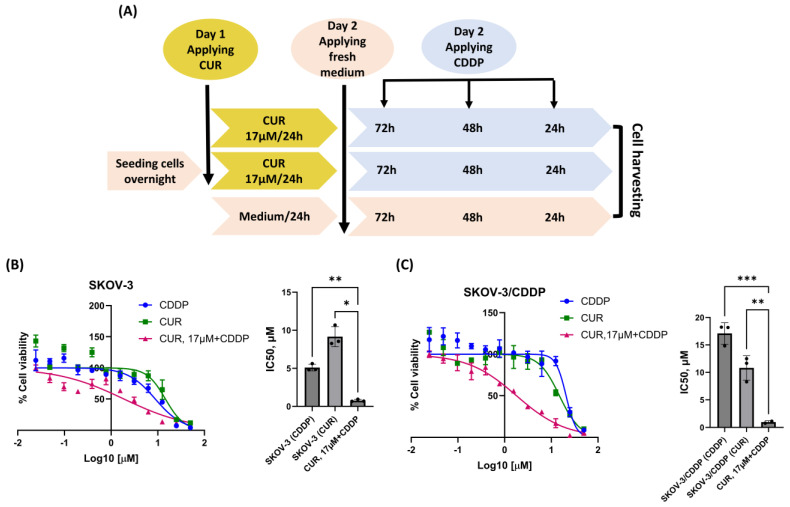
Scheme of the suggested strategy used in this study (**A**). Effect of CUR pre-treatment followed by CDDP on the viability of ovarian cancer cells by MTT assay. Both (**B**, **left**) SKOV-3 and (**C**, **left**) SKOV-3/CDDP cancer cell lines were treated with either CUR or CDDP for 72 h or 17 μM CUR for 24 h followed by serial concentrations of CDDP for another 72 h. Data plotted are the mean ± SEM, *, *p* < 0.05; **, *p* < 0.01; ***, *p* < 0.001, ANOVA test for (**B**, **right**) SKOV-3 and (**C**, **right**) SKOV-3/CDDP.

**Figure 3 ijms-26-00799-f003:**
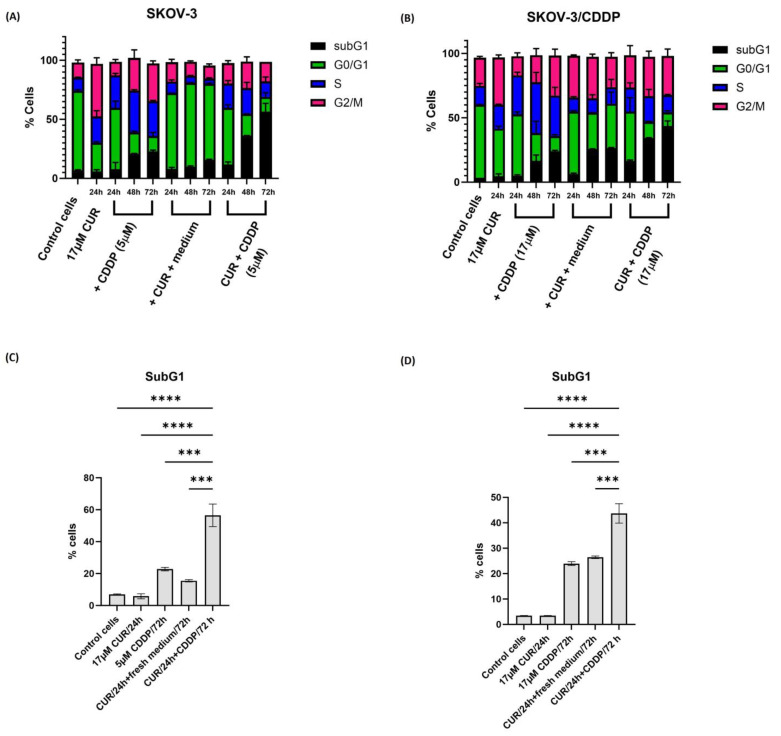
Cell cycle analysis by flow cytometry in ovarian (**A**) SKOV-3 and (**B**) SKOV-3/CDDP cancer cell lines treated with either (1/2 IC_50_, 5 µM, or 17 µM for SKOV-3 and SKOV-3/CDDP cell lines, respectively) CDDP for 24 h, 48 h, and 72 h alone or (17 µM) CUR for 24 h. Then, the medium was replaced with a fresh culture medium for 24 h, 48 h, and 72 h alone or CUR pre-treatment by treatment firstly with (17 µM) CUR for 24 h followed by treatment with CDDP for another 24 h, 48 h, and 72 h (5 µM or 17 µM for SKOV-3 and SKOV-3/CDDP cell lines, respectively). The percentage of cells in the sub-G1 phase significantly increased in CUR pre-treatment compared to either CDDP or CUR groups. Data plotted are the mean ± SEM, ***, *p* < 0.001; **** *p* < 0.0001, ANOVA test for (**C**) SKOV-3 and (**D**) SKOV-3/CDDP cells.

**Figure 4 ijms-26-00799-f004:**
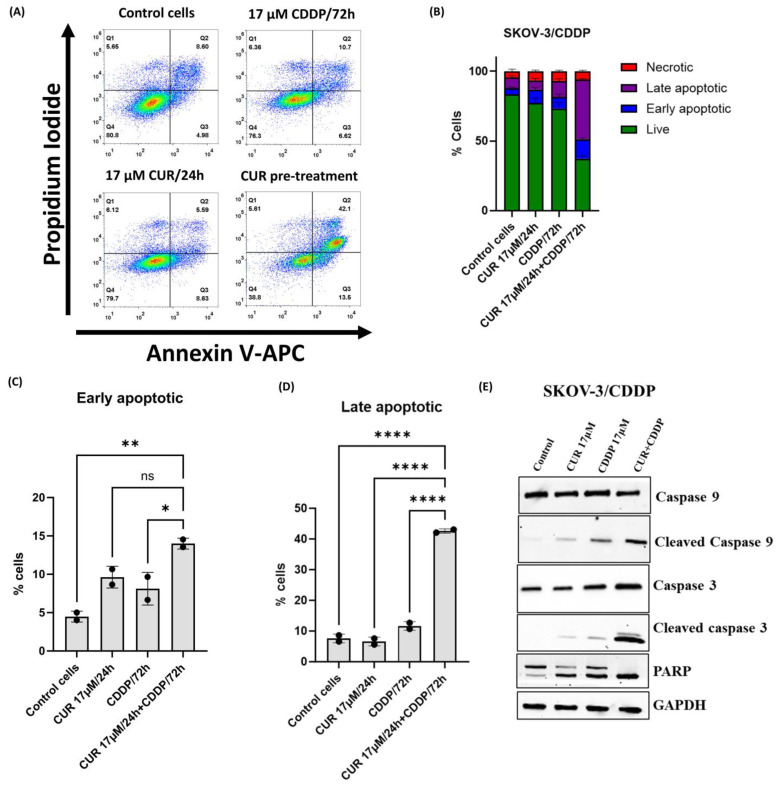
Flow cytometric analysis of ovarian SKOV-3/CDDP cancer cells after treatment with 17 μM CUR for 24 h alone, 17 μM CDDP for 72 h alone, or 17 μM CUR for 24 h, followed by the treatment of CDDP for another 72 h (CUR pre-treatment). Cells were stained with Annexin V-FITC and PI solution and analyzed with flow cytometry. (**A**) Plots and (**B**) bar graph show that the proportion of apoptotic cells in the early and late stages is increased in CUR pre-treatment compared to single treatment groups. Data plotted are the mean ± SEM, *, *p* < 0.05; **, *p* < 0.01; **** *p* < 0.0001, ns means not significant, ANOVA test for (**C**) early and (**D**) late apoptosis. (**E**) Apoptosis-related protein expression levels of cleaved caspase 9, cleaved caspase 3, and cleaved PARP in SKOV-3/CDDP cells were analyzed by western blotting.

**Figure 5 ijms-26-00799-f005:**
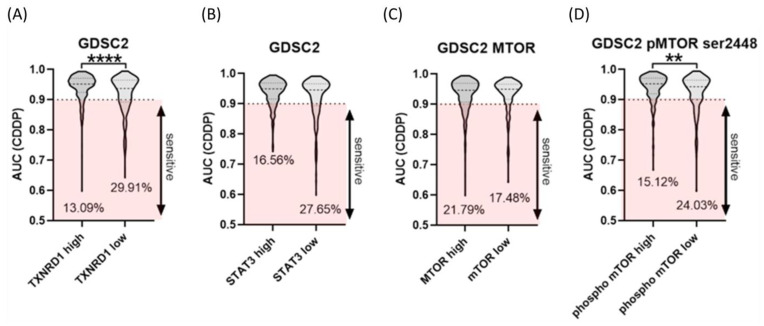
Violin plots comparing the four included datasets, including the number of investigated agents. (**A**) *TXNRD1* gene. (**B**) *STAT3* gene. (**C**) mTOR and (**D**) mTOR p-Ser^2448^ proteins. Data plotted are the mean ± SEM, **, *p* < 0.01; **** *p* < 0.0001, Mann–Whitney test.

**Figure 6 ijms-26-00799-f006:**
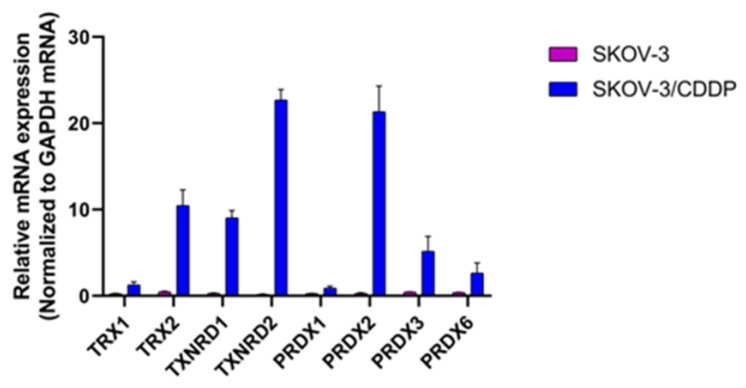
Relative mRNA expression levels of thioredoxin system genes as determined by real-time quantitative RT-PCR.

**Figure 7 ijms-26-00799-f007:**
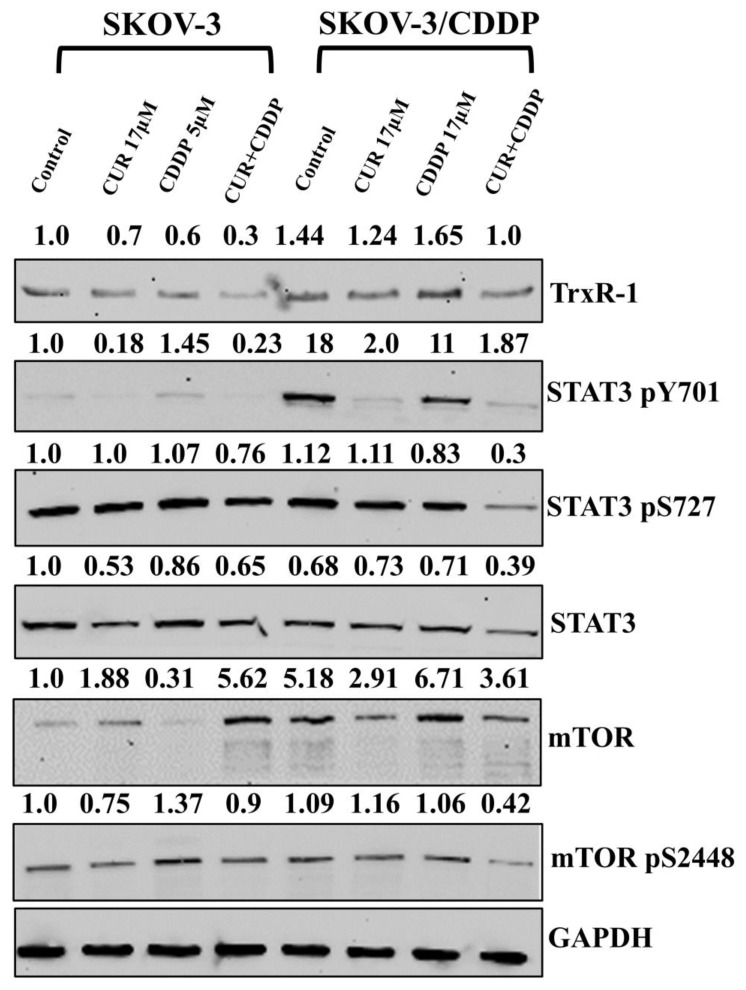
The protein levels of TRXR1, STAT3, p-STAT3, mTOR and p-mTOR were detected using Western blot. GAPDH was used as an internal control. Data analysis showed that the CUR pre-treatment inhibits TRXR1 and mTOR/ STAT3 signaling pathways in SKOV-3/CDDP cells compared to mono-treatment groups. Ovarian SKOV-3/CDDP cells were treated with (17 μM) CUR or (5 µM and 17 µM for 24 h and 48 h for SKOV-3 and SKOV-3/CDDP, respectively) CDDP alone or 17 μM CUR for 24 h followed by treatment of CDDP for another 24 h for SKOV-3 or 48 h for SKOV-3/CDDP (CUR pre-treatment).

**Table 1 ijms-26-00799-t001:** List of primers used for real-time RT-PCR.

Gene	Sense 5′–3′	Antisense 5′–3′
*TRX1*	TGGTGAAGCAGATCGAGAGCAAGA	ACCACGTGGCTGAGAAGTCAACTA
*TRX2*	TGGTGGCCTGACTGTAACAC	TGTTGACCACTCGGTCTTGA
*TXNRD1*	GTGTTGTGGGCTTTCACGTA	TGGTCAGTCCACATTTGAGC
*TXNRD2*	GCCATAGCACCTTGCATCTC	ATCCTCGATGAGGACACCTG
*PRDX1*	ACAGCCGTTGTCAATGGAGAG	ACGTCGTGAAATTCGTTAGCTT
*PRDX2*	CTGGCGAAGGACACCCTTGCCATC	GGCCACAGCGGTGGTTGATGGCG
*PRDX3*	CTTGGTGTATTTATCCAGGCAAGATGGC	GGCCTGCTGCATGTGGAAGAACGA
*PRDX6*	CAACTTTGAGGCCAATACCA	CAACTTAACATTCCTCTTGG
*GAPDH*	GAAGGTGAAGGTCGGAGTC	GAAGATGGTGATGGGATTTC

## Data Availability

Data are contained within the article and Appendix A.

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
