# Peer review of "Re-Sensitization of Resistant Ovarian Cancer SKOV3/CDDP Cells to Cisplatin by Curcumin Pre-Treatment"

_ijms, 2025, doi:10.3390/ijms26020799_

Round 1

Reviewer 1 Report

Comments and Suggestions for Authors

Major Comments:

1. In Figure 3, the results are difficult to interpret because the font size of the image is too small. Could you provide a representative bar graph or enlarge the image size? Based on the results in the figure, it seems that the parallel treatment of CDDP and CUR has no significant effect compared to single-agent treatments in the cell cycle. Also, these data need to be analyzed using ANOVA.

2. In the current FACs data of figure 4, the representation of apoptotic cells appears unclear or partially obscured. Please adjust the data visualization to clearly show the apoptotic cell populations, particularly in the Annexin V+/PI+ quadrant. "Furthermore, the data should be analyzed using ANOVA to ensure statistical significance. In Figure 4, why are normal SKOV-3 cells not included?

3. Is the purpose of this paper to focus on the synergistic effect of CUR and CDDP on CDDP-resistant cells rather than on the effect on general cancer cells? I think it would have no effect on general cancer cells and would only show limited effects when applied to resistant cancer cells.

4. The results from apoptosis analysis (Result 4) and TrxR1 expression (Result 5) are presented separately, but their mechanistic relationship is unclear. Please elaborate on how the suppression of TrxR1 by CUR-CDDP treatment contributes to the induction of apoptosis.

5. The explanation for why combined treatment with curcumin and cisplatin shows an antagonistic effect is insufficient. In particular, the molecular and cellular reasons for the reduced effect of co-administration of the two drugs are not specifically discussed.

6. The legend describing Figure 3 looks weird, so it would look better if sentences were modified. Moreover, the percentage of cell phase is written with too small font size, so I’m not sure if the Sub-G1 phase has been increased or not for real in the pre-treatment group. Also, is there any reason that cell cycle assay has been performed at different times? (SKOV-3: 24h, 48h, SKOV-3/CDDP: 48h, 72h)

7. In figure 5, (A), (B), (C) has been deleted. Also, the discussion about the meaning of figure 5 seems insufficient. -----> For instance, what is the meaning of the percentage in figure exactly? Furthermore, authors explained that ‘over-expression of TRXR1 genes and increased levels of mTOR p-Ser2448 protein is statistically significant correlated with resistance against CDDP in tested cancer cell lines’, but according to figure, isn’t that represent opposite pattern each other?

8. To clearly assert that pretreatment of CUR inhibits the TrxR1 and mTOR/STAT3 signaling pathway, additional results representing relative band intensity ratio or demonstrating statistical significance need to be added. (For example, the expression of TrxR-1, STAT3 pY705, STAT3, mTOR doesn’t look so different upon pretreatment of CUR)

9. The paper highlights the effects of curcumin by inhibiting **thioredoxin reductase (TrxR)** and the mTOR/STAT3 pathway. However, there is a lack of mechanistic explanation as to how these pathway inhibitions specifically reduce cisplatin resistance in cells. For example, there is a lack of discussion on whether inhibition of TrxR and downregulation of the mTOR/STAT3 pathway contribute to additional anticancer mechanisms other than apoptosis induction.

10. When the CUR and CDDP were co-treated to ovarian cancer cells, it was one of the important discoveries that the antagonist effect has occurred, but there is a lack of evidence or discussion about “Why did the antagonist effect occur?” ----> It seems like further explanation is needed. In addition, “Based on the preclinical studies, polyphenols have been shown to work synergistically with chemotherapeutics drugs, but some polyphenols work antagonistically [33]” ----> It seems not enough to assert that the antagonist effect can occur with this one paper.

Minor Comments:

1. In Result 2.1, please describe the IC50 concentrations of CUR and CDDP used in the experiments.

2. Please provide detailed information about the parameters on the X-axis and Y-axis in Figures 1A and 1B.

3. In Figure 2, the data need to be analyzed using ANOVA, and the p-values should be reported.

4. In 176 lines, there is a typo, ‘CDDD’ should be corrected to ‘CDDP’.

5. To better represent the progression of apoptosis, the Western blot data in Figure 4B should be reorganized based on the activation sequence of key apoptotic markers. The recommendation is: Caspase-9 activation ----> Caspase-3 activation ---->PARP cleavage.

Author Response

First of all, we would like to thank reviewers for their comments and suggestions that allowed us to greatly improve the quality of the manuscript. We agree with all comments, and we corrected point by point the manuscript accordingly. The comments of the reviewers are in bold text and our responses in plain italics.

Author's Reply to the Review Report (Reviewer 1)

In Figure 3, the results are difficult to interpret because the font size of the image is too small. Could you provide a representative bar graph or enlarge the image size? Based on the results in the figure, it seems that the parallel treatment of CDDP and CUR has no significant effect compared to single-agent treatments in the cell cycle. Also, these data need to be analyzed using ANOVA.

            Figure 3 has been modified (page #7, lines#228-229). The plot has been moved to the supplementary (Figure S1), and the bar chart graph been inserted so that the effect of CUR with CDDP is more clearly seen compared to the single treatment. The data has been analyzed using ANOVA with correction for multiple comparisons.        

In the current FACs data of figure 4, the representation of apoptotic cells appears unclear or partially obscured. Please adjust the data visualization to clearly show the apoptotic cell populations, particularly in the Annexin V+/PI+ quadrant. "Furthermore, the data should be analyzed using ANOVA to ensure statistical significance. In Figure 4, why are normal SKOV-3 cells not included?

Figure 4 has been improved (page #8, lines#255-256). The bar graph has been added. The data has been analyzed using ANOVA.

In figure 3 we approved that the CUR and CDDP treatment activated the apoptotic cells percentage (sub G1) in normal SKOV-3 cells. Since normal cells are sensitive to CDDP, it is obvious that treatment with CUR and CDDP would induce cell death. Our study aims to highlight the role of CUR and CDDP together in increasing the sensitivity of cisplatin-resistant SKOV-3/CDDP cells to CDDP and to avoid distracting the reader's mind to look at the sensitive cells (normal SKOV-3). Especially since the results of protein expression in this study showed that treatment with CUR and CDDP did not lead to a decrease in the expression of proteins associated with the acquisition of CDDP resistance in these sensitive cells. Therefore, the results related to resistant cells were discussed later without delving into the interpretation of the results related to sensitive SKOV-3 cells.

Is the purpose of this paper to focus on the synergistic effect of CUR and CDDP on CDDP-resistant cells rather than on the effect on general cancer cells? I think it would have no effect on general cancer cells and would only show limited effects when applied to resistant cancer cells.

The purpose of this study is to study the effect of CUR-CDDP treatment on resensitizing cisplatin-resistant cells to cisplatin. The results using the DEPMAP database have shown that the mechanism of resistance to cisplatin in most cancer cells in general is due to increased expression of thioredoxin reductase and phosphorylation of mTOR protein. Our results have also shown that treatment of cancer cells with curcumin and cisplatin led to increased sensitivity of cells to cisplatin after decreased expression of these proteins.

The following sentence was written in advance in the discussion section (12, lines#358-361).

Despite extensive studies have shown that the synergistic role of CUR in enhancing the efficacy of the most common chemotherapy drugs, its potential therapeutic role in ovarian cancer is still need to elucidate, especially on SKOV-3 cells.

The main purpose of this paper was to show the synergistic effect of CUR and CDDP on CDDP-resistant ovarian cells and investigate the mechanism of resensitization. Unexpectedly, these cells showed no response to the synergistic effect, which made us think of a new strategy to resensitize cells to cisplatin.

Using an alternative strategy, the study showed that pretreatment actually increased the cells' sensitivity to cisplatin. Whether this strategy (pretreatment) is effective on other cells or not must be tested practically and then decided. It is possible to develop this study later to include other cancer cells resistant to cisplatin and make a comparison between them.

The results from apoptosis analysis (Result 4) and TrxR1 expression (Result 5) are presented separately, but their mechanistic relationship is unclear. Please elaborate on how the suppression of TrxR1 by CUR-CDDP treatment contributes to the induction of apoptosis.

We completely agree with you. A new paragraph has been added in the discussion section (page #14, lines 464-478) to elaborate on how the suppression of TrxR1 by CUR-CDDP treatment contributes to the induction of apoptosis. Three new references have been cited.

The reduced form of thioredoxin catalyzes reduction of disulfide bonds in target proteins, including STAT3. Elevated levels of Thioredoxin reductase and STAT3 signaling have been associated with the development of chemotherapy resistance (Figure 5, 6 and 7). In cancer cells, the inhibition of TrxR1 TXNRD expression results in increased oxidative stress and the accumulation of oxidized Prx2 and STAT3, which blocks STAT3-dependent transcription. In this study after CUR-CDDP treatment, the inhibition of TrxR1 TRXR1after CUR-CDDP treatment results in the ac-cumulation of oxidized STAT3, which blocks STAT3-dependent transcription. Targeting STAT3 signaling by CUR-CDDP could interrupt the anti-apoptotic function of ovarian cancer cells that lead to increase apoptotic cell death. Furthermore, activation of mTORC1 leads to direct Ser727 phosphorylation of STAT3 during hypoxia and promoting hypoxia inducible factors (HIF-1α) mRNA transcription and vascular endothelial growth factors (VEGF-A). Thus, our results suggested that CUR pre-treatment could inhibit mTOR phosphorylation. This, in turn, leads to a decrease of phosphorylation in Ser727 in STAT3 to confer re-sensitization to CDDP drugs inSKOV-3/CDDP cells.

The explanation for why combined treatment with curcumin and cisplatin shows an antagonistic effect is insufficient. In particular, the molecular and cellular reasons for the reduced effect of co-administration of the two drugs are not specifically discussed.

Many articles proved that the combination treatment of curcumin and cisplatin is synergistic in cancer cells, and there has been no evidence of antagonistic effect between curcumin and cisplatin. Since the co-treatment has not been studied in this type of ovarian cells, in our study, the results of the toxicity test and cell cycle showed that the co-treatment has an antagonistic effect on cisplatin resistant in SKOV-3 ovarian cells.

We agree with you. A new paragraph has been added in the discussion section (page # 12 and 13, lines 392-404) to give some cellular reasons for the reduced effect of co-administration of the two drugs. Three new references have been cited.

The strong antagonistic interaction between polyphenols and CDDP has been demonstrated in several studies in kidney cases. Due to strong antioxidative activities, polyphenols could prevent cisplatin-induced damage by scavenging ROS. For example, some polyphenols, including green tea polyphenol, epicatechin gallate (ECG), prevented cisplatin-induced oxidative stress, inflammation, and apoptosis by downregulating the MAPK pathway, leading to improved renal function in cisplatin-treated rat. The antagonistic effect can be explained by one drug targets cells at different phases of the cell cycle, when one drug prevents the progression of the cell cycle to a phase where the second drug has its effect making the combination treatment less active than mono-treatment. In our case blocking of the cells at G2/M phase by CUR, prevents further damage in the replication phase by CDDP, while the washout of the drug increases the susceptibility of damaged cells to the CDDP in the subsequent round of replication

The legend describing Figure 3 looks weird, so it would look better if sentences were modified. Moreover, the percentage of cell phase is written with too small font size, so I’m not sure if the Sub-G1 phase has been increased or not for real in the pre-treatment group. Also, is there any reason that cell cycle assay has been performed at different times? (SKOV-3: 24h, 48h, SKOV-3/CDDP: 48h, 72h)

            Figure 3 has been modified. Its size has been enlarged and the plot has been moved to the supplementary (Figure S1). The bar chart graph been included so that the effect of CUR with CDDP is more clearly seen compared to the single treatment after analyzing the data using ANOVA (page #7, lines#228-229).   

Cell cycle assay was performed at different times for both SKOV-3 and SKOV-3/CDDP at 24h, 48h, and 72h but after 24 h of single treatment or co-treatment no change was seen in the resistant cells when compared to control cells, so this statement was written in advance. (For SKOV-3/CDDP cell line there was little change in survival until approximately 48h). (page #6, lines#221-222).

In figure 5, (A), (B), (C) has been deleted. Also, the discussion about the meaning of figure 5 seems insufficient. -----> For instance, what is the meaning of the percentage in figure exactly? Furthermore, authors explained that ‘over-expression of TRXR1 genes and increased levels of mTOR p-Ser2448 protein is statistically significant correlated with resistance against CDDP in tested cancer cell lines’, but according to figure, isn’t that represent opposite pattern each other?

Numbers have been added to the figure (page #9, lines#289-290).

The percentages represent the number of cell lines in the DEPMAP database which we classified as “sensitive” (with AUC for cytotoxicity graphs <0.9). The cell lines were divided into two groups – with low and high level of the marker (less than median, and more than median of expression/phosphorylation level). For both TXNRD1 expression (A) and mTOR phosphorylation on ser2448 (D), the number of “sensitive” cell lines was lower in the “high expression” group. The same is not true for total mTOR expression and STAT3 expression. We tried to clarify these points in the manuscript:

A new paragraph has been added (page #9, lines#277-288) containing percentages and explaining at the same time that it appears that overexpression of TRXR1 genes and increased levels of mTOR p-Ser2448 protein are statistically significantly associated with resistance against CDDP in the tested cancer cell lines, and that they do not represent opposite pattern each other.

As shown in Figure 5, analysis of cell lines in the DEPMAP showed that high levels (> median) of TXNRD1 gene and increased levels of mTOR p-Ser2448 protein are statistically significantly correlated to resistance against CDDP in tested cancer cell lines. The number of cancer cell lines with low expression of TXNRD1 gene and mTOR p-Ser2448 protein that show a sensitivity (AUC<0.9) to CDDP (29.9% and 24.03%) was higher compared to the  cell lines with high expression of TRXR1 and mTOR p-Ser2448 (13.09% and 15.12% of cell lines) that are sensitive to CDDP. Although the number of resistant cell lines with increased levels of STAT3 was higher, this difference was not statistically significant.

To clearly assert that pretreatment of CUR inhibits the TrxR1 and mTOR/STAT3 signaling pathway, additional results representing relative band intensity ratio or demonstrating statistical significance need to be added. (For example, the expression of TrxR-1, STAT3 pY705, STAT3, mTOR doesn’t look so different upon pretreatment of CUR)

Figure 7 has been modified, containing density measurement (page #11, lines#326-327). At the protein level, the treatment does not show any change in the percentage of in inactive STAT3 and m-TOR proteins (un-phosphorylated forms), at the level of the active or phosphorylated p-STAT3 ser and p-mTOR proteins, a decrease in expression was observed.

The paper highlights the effects of curcumin by inhibiting **thioredoxin reductase (TrxR)** and the mTOR/STAT3 pathway. However, there is a lack of mechanistic explanation as to how these pathway inhibitions specifically reduce cisplatin resistance in cells. For example, there is a lack of discussion on whether inhibition of TrxR and downregulation of the mTOR/STAT3 pathway contribute to additional anticancer mechanisms other than apoptosis induction.

A new paragraph has been added in the discussion section (page #14, lines 464-478) to give some explanation of how the inhibition of these proteins contribute to anticancer mechanisms lead to resensitization of SKOV-3/CDDP to CDDP. Three new references have been cited.

The reduced form of thioredoxin catalyzes reduction of disulfide bonds in target proteins, including STAT3. Elevated levels of Thioredoxin reductase and STAT3 signaling have been associated with the development of chemotherapy resistance (Figure 5, 6 and 7). In cancer cells, the inhibition of TrxR1 TXNRD expression results in increased oxidative stress and the accumulation of oxidized Prx2 and STAT3, which blocks STAT3-dependent transcription. In this study after CUR-CDDP treatment, the inhibition of TrxR1 TRXR1after CUR-CDDP treatment results in the ac-cumulation of oxidized STAT3, which blocks STAT3-dependent transcription. Targeting STAT3 signaling by CUR-CDDP could interrupt the anti-apoptotic function of ovarian cancer cells that lead to increase apoptotic cell death. Furthermore, activation of mTORC1 leads to direct Ser727 phosphorylation of STAT3 during hypoxia and promoting hypoxia inducible factors (HIF-1α) mRNA transcription and vascular endothelial growth factors (VEGF-A). Thus, our results suggested that CUR pre-treatment could inhibit mTOR phosphorylation. This, in turn, leads to a decrease of phosphorylation in Ser727 in STAT3 to confer re-sensitization to CDDP drugs inSKOV-3/CDDP cells.

When the CUR and CDDP were co-treated to ovarian cancer cells, it was one of the important discoveries that the antagonist effect has occurred, but there is a lack of evidence or discussion about “Why did the antagonist effect occur?” ----> It seems like further explanation is needed. In addition, “Based on the preclinical studies, polyphenols have been shown to work synergistically with chemotherapeutics drugs, but some polyphenols work antagonistically [33]” ----> It seems not enough to assert that the antagonist effect can occur with this one paper.

We agree with you. A new paragraph has been added in the discussion section (page # 12 and 13, lines 392-404) to give some explanation. Three new references have been cited.

The strong antagonistic interaction between polyphenols and CDDP has been demonstrated in several studies in kidney cases. Due to strong antioxidative activities, polyphenols could prevent cisplatin-induced damage by scavenging ROS. For example, some polyphenols, including green tea polyphenol, epicatechin gallate (ECG), prevented cisplatin-induced oxidative stress, inflammation, and apoptosis by downregulating the MAPK pathway, leading to improved renal function in cisplatin-treated rat. The antagonistic effect can be explained by one drug targets cells at different phases of the cell cycle, when one drug prevents the progression of the cell cycle to a phase where the second drug has its effect making the combination treatment less active than mono-treatment. In our case blocking of the cells at G2/M phase by CUR, prevents further damage in the replication phase by CDDP, while the washout of the drug increases the susceptibility of damaged cells to the CDDP in the subsequent round of replication.  

In Result 2.1, please describe the IC50 concentrations of CUR and CDDP used in the experiments

          A paragraph has been changed in the materials and methods section (page # 15, lines 507-509). IC50 values for CUR (17µM and 47µM for SKOV-3 and SKOV-CDDP, respectively) and CDDP (10µM and 34µM for SKOV-3 and SKOV-3/CDDP cells, respectively) were established in our previous papers.

Please provide detailed information about the parameters on the X-axis and Y-axis in Figures 1A and 1B.

A new paragraph has been added under figure 1 (page # 4, lines 180-182). The Y axis, fraction affected (Fa) represents the ratio of the combination effect on inhibition of cell growth; the X axis represents the doses of drugs.

In Figure 2, the data need to be analyzed using ANOVA, and the p-values should be reported.

In 176 lines, there is a typo, ‘CDDD’ should be corrected to ‘CDDP’.

In Figure 2, the data has been analyzed using ANOVA (page #5, line#206-207) and p-values has been reported.

The typo has been corrected (page #4, line#174).

To better represent the progression of apoptosis, the Western blot data in Figure 4B should be reorganized based on the activation sequence of key apoptotic markers. The recommendation is: Caspase-9 activation ----> Caspase-3 activation ---->PARP cleavage.

Figure 4 has been improved ((page #8, lines#255-256).

We would like to sincerely thank you for your advices and constructive comments.

Sincerely,

Hasan Aseel on behalf of all the authors

Reviewer 2 Report

Comments and Suggestions for Authors

This study investigated the anti-tumor effects of curcumin (CUR) on a cisplatin (CDDP)-resistant ovarian cancer cell line (SKOV3/CDDP). The authors demonstrated that CUR pre-treatment, but not concomitant treatment, synergistically sensitized the resistant cells to CDDP. Moreover, they obtained some mechanistic insights of this effects by determining mRNA and protein expression levels. These data are clear and robust, but preliminary to support their conclusion. I would like the authors to address the following concerns to draw a firm conclusion.

1. Introduction is too long, should more focus on specific topics related to the present study, such as therapeutic effects and clinical concerns of CDDP, anti-tumor effects of CUM, and fundamental mechanisms of CDDP resistance involving antioxidative enzymes.

2. Figure 3:

a) The experimental procedure is difficult to be understood. A time flow chart of the experimental design is needed.

b) Different incubation periods between SKOV3 and SKOV3/CDDP should be rationalized.

c) The resolution of the images is too low to see.

d) A chart summarizing cell cycle distribution should be provided.

3. Figure 4: Region Q2 represents late apoptotic/necrotic cells; should not be used to discuss apoptosis. Instead, the frequency of region Q3 representing early apoptosis should be used.

4. TXNRD1 is the official gene symbol, pleas avoid TRXR1, TRXR, TrxR-1 throughout the manuscript.

5. Figures 5–7: These results are observational and do not indicate any causal relationships. It should be demonstrated that pharmacological inhibition or knockdown of one of these factors enhance the sensitivity of SKOV3/CDDP cells to CDDP.

Author Response

First of all, we would like to thank reviewers for their comments and suggestions that allowed us to greatly improve the quality of the manuscript. We agree with all comments, and we corrected point by point the manuscript accordingly. The comments of the reviewers are in bold text and our responses in plain italics.

Author's Reply to the Review Report (Reviewer 2)

Introduction is too long, should more focus on specific topics related to the present study, such as therapeutic effects and clinical concerns of CDDP, anti-tumor effects of CUM, and fundamental mechanisms of CDDP resistance involving antioxidative enzymes.

The introduction has been shortened to fit the study. Two paragraphs have been deleted (page # 2, lines 62-84) and (page # 3, lines 121-129).

Figure 3:

  1. a) The experimental procedure is difficult to be understood. A time flow chart of the experimental design is needed.

The experimental scheme has been inserted at the beginning of the study in figure 2A (page #5, line#200-201).

New paragraph has been inserted (page # 5, lines 188-189).

 To determine whether CUR sensitizes SKOV-3/CDDP cell line to CDDP, a suggested CUR pre-treatment therapeutic strategy was designed (Figure 2A).

  1. b) Different incubation periods between SKOV3 and SKOV3/CDDP should be rationalized.

            Figure 3 has been improved. Incubation periods between SKOV3 and SKOV3/CDDP have been rationalized (page #7, lines#228-229).

  1. c) The resolution of the images is too low to see.

Figure 3 has been improved (page #7, lines#228-229). The plot has been moved to the supplementary after modification (Figure S1).

  1. d) A chart summarizing cell cycle distribution should be provided.

Figure 3 has been improved. The bar chart graph has been included (page #7, lines#228-229).

Figure 4: Region Q2 represents late apoptotic/necrotic cells; should not be used to discuss apoptosis. Instead, the frequency of region Q3 representing early apoptosis should be used.

            We agree with you. The paragraph has been changed in the result section (page # 7, lines 244-245). However, the CUR pre-treatment increased the percentage of early and total apoptotic cells to 13.5% and 42% at 72 h, respectively (Figure 4A, B, C, D).

            Figure 4 has been improved (page #8, lines#255-256). The bar graph for early and late apoptosis has been added.

  1. TXNRD1 is the official gene symbol, pleas avoid TRXR1, TRXR, TrxR-1 throughout the manuscript.

            The abbreviations have been replaced by the official gene (TXNRD1) and protein (TRXR1) symbols throughout the manuscript.

  1. Figures 5–7: These results are observational and do not indicate any causal relationships. It should be demonstrated that pharmacological inhibition or knockdown of one of these factors enhance the sensitivity of SKOV3/CDDP cells to CDDP.

The study was designed based on our previous data and data published by other groups. A new paragraph has been added in the discussion section (page #14, lines 464-478) to elaborate on how the suppression of TrxR1 by CUR-CDDP treatment contributes to the induction of apoptosis. Three new references have been cited.

The reduced form of thioredoxin catalyzes reduction of disulfide bonds in target proteins, including STAT3. Elevated levels of Thioredoxin reductase and STAT3 signaling have been associated with the development of chemotherapy resistance (Figure 5, 6 and 7). In cancer cells, the inhibition of TrxR1 TXNRD expression results in increased oxidative stress and the accumulation of oxidized Prx2 and STAT3, which blocks STAT3-dependent transcription. In this study after CUR-CDDP treatment, the inhibition of TrxR1 TRXR1after CUR-CDDP treatment results in the ac-cumulation of oxidized STAT3, which blocks STAT3-dependent transcription. Targeting STAT3 signaling by CUR-CDDP could interrupt the anti-apoptotic function of ovarian cancer cells that lead to increase apoptotic cell death. Furthermore, activation of mTORC1 leads to direct Ser727 phosphorylation of STAT3 during hypoxia and promoting hypoxia inducible factors (HIF-1α) mRNA transcription and vascular endothelial growth factors (VEGF-A). Thus, our results suggested that CUR pre-treatment could inhibit mTOR phosphorylation. This, in turn, leads to a decrease of phosphorylation in Ser727 in STAT3 to confer re-sensitization to CDDP drugs inSKOV-3/CDDP cells.

Thank you for your suggestion. We completely agreed with you, will use pharmacological inhibition or knockdown of one of these factors enhance the sensitivity of SKOV3/CDDP cells to CDDP later on in our further studies.

We would like to sincerely thank you for your advices and constructive comments.

Sincerely,

Hasan Aseel on behalf of all the authors

Round 2

Reviewer 2 Report

Comments and Suggestions for Authors

The authors addressed my comments. I have no further concerns.